# Genome-Wide Identification and Co-Expression Networks of *WOX* Gene Family in *Nelumbo nucifera*

**DOI:** 10.3390/plants13050720

**Published:** 2024-03-04

**Authors:** Juan-juan Li, Xiao-yan Qiu, Yu-jun Dai, Tonny M. Nyonga, Chang-chun Li

**Affiliations:** 1Hubei Province Research Center of Engineering Technology for Utilization of Botanical Functional Ingredients, Hubei Key Laboratory of Quality Control of Characteristic Fruits and Vegetables, College of Life Science and Technology, Hubei Engineering University, Xiaogan 432000, China; juanjuan_black@hbeu.edu.cn (J.-j.L.); qiuxiaoyan0403@163.com (X.-y.Q.); dyj5925@hbeu.edu.cn (Y.-j.D.); 2Department of Biology, University of New Mexico, Albuquerque, NM 87131, USA; maragatonny@outlook.com

**Keywords:** lotus, phylogenetic analysis, expression pattern, co-expression network

## Abstract

WUSCHEL-related homeobox (*WOX*) genes are a class of plant-specific transcription factors, regulating the development of multiple tissues. However, the genomic characterizations and expression patterns of WOX genes have not been analyzed in lotus. In this study, 15 *NnWOX* genes were identified based on the well-annotated reference genome of lotus. According to the phylogenetic analysis, the *NnWOX* genes were clustered into three clades, i.e., ancient clade, intermediate clade, and WUS clade. Except for the conserved homeobox motif, we further found specific motifs of *NnWOX* genes in different clades and divergence gene structures, suggesting their distinct functions. In addition, two *NnWOX* genes in the ancient clade have conserved expression patterns and other *NnWOX* genes exhibit different expression patterns in lotus tissues, suggesting a low level of functional redundancy in lotus *WOX* genes. Furthermore, we constructed the gene co-expression networks for each *NnWOX* gene. Based on weighted gene co-expression network analysis (WGCNA), ten *NnWOX* genes and their co-expressed genes were assigned to the modules that were significantly related to the cotyledon and seed coat. We further performed RT-qPCR experiments, validating the expression levels of ten *NnWOX* genes in the co-expression networks. Our study reveals comprehensive genomic features of *NnWOX* genes in lotus, providing a solid basis for further function studies.

## 1. Introduction

*WUSCHEL*-related homeobox (*WOX*) genes are a plant-specific transcription factor family, belonging to the homeobox superfamily [1,2]. The proteins of *WOX* genes contain a common homeobox domain, a helix–turn–helix structure consisting of 60–66 amino acids, which was a key regulatory element recognized by specific DNA sequences and regulated target gene expression at different stages of plant development [3,4]. According to most phylogenetic analysis, *WOX* genes were divided into three separate clades, modern/WUS clade (WC), intermediate clade (IC), and ancient clade (AC) [5]. The gradual expansion trend of the *WOX* gene number was approximately matched with the continuous evolution from lower plants to higher plants. Additionally, the last common ancestors of *WOX* proteins were identified in green algae [6].

Recent genetics and molecular biology investigations of *WOX* genes have indicated the vital roles of *WOX* members in various physiological and developmental processes in plants [7,8,9]. Since Vollbrecht et al. first found the *Knotted-1* gene (i.e., *WOX* gene) that contained a homeodomain contributed to the cell fate determination in a maize mutant [10], an increasing number of studies have focused on the biological functions of *WOX* members in different plant tissues. In particular, a total of 15 *WOX* genes have been identified in the model eudicot plant *Arabidopsis thaliana* [11,12]. Several *AtWOX* genes are essential for multiple processes of tissue formation, including *AtWUS* and *PRS1/AtWOX3* in the shoot apical meristem (STM) [13,14], *AtWOX2* in the egg cell and zygote [15], *AtWOX6* in the ovule development [16], *STIP/AtWOX9* in STM, the root and other aerial organs [17], and *AtWOX13* in floral transition [18]. In a model rice plant, the interaction of *OsWOX11* and *SLG2* determined grain width and affected quality traits [19]. Variation in the gene expression level of *OsWUS* negatively affects the phenotypic traits in rice [20]. The expression patterns of *OsWOX* genes were significantly divergent in reproductive organs, vegetative tissues, and multiple tissue-specific expressed *OsWOX* genes where they were identified [21]. The genome-wide identifications of the *WOX* gene family in non-model plants were conducted on various crops [22,23,24,25,26]. Furthermore, due to the changing expression level, *WOX* genes help plants respond to abiotic stress, indicating a more complex regulatory network of *WOX* genes under environmental stress. For example, a complex consisting of *WOX* and *FRUCTOSE INSENSITIVE1* proteins regulated the expression level of downstream genes where *Arabidopsis* responds to fructose signaling during growth and development [27]. A reduction in *WOX* gene expression in the roots of tobacco seedlings grown under induction of alanine–glutamine acid–asparagine acid–leucine increases the differentiation of stem cells and leads to root elongation [28]. However, only a few studies have constructed the gene regulatory network of *WOX* genes, which limits further exploration of the complex interactions of *WOX* genes in plants.

*Nelumbo nucifera*, also called sacred lotus, is a considerable aquatic crop widely planted in Asia. With specific purposes of artificial domestication, the cultural lotuses were divided into three taxonomic groups, i.e., rhizome, seed, and flower lotus [29]. Based on the well-assembled reference genome of lotus [30], recent transcriptomic and proteomic research on sacred lotus was carried out to reveal the molecular mechanism of regulating the development of tissues [31,32,33]. According to previous studies, a single whole-genome duplication event occurred in ancestors of the *N. nucifera* during the K-Pg boundary [30], similar to the basal eudicot *Amborella trichopoda* [34]. Notably, the *Nelumbo* genome database including multiple tissue RNA-seq samples provided a superb platform for lotus investigation [35]. Several transcription factor families were identified in lotus, such as MADS-box [36], *WRKY* [37], and *YABBY* [38]. Although a previous study conducted a preliminary analysis of the WOX gene family in lotus [39], the interactions of *WOX* genes is still largely unknown. Therefore, we describe *WOX* genes systematically in *N. nucifera* and construct the co-expressed networks to find vital interactions regarding *NnWOX* genes during plant growth and development. This study consists of several parts, including the identification of *WOX* family members in *N. nucifera*, phylogenetic tree analysis, chromosomal localization, gene structure, conserved domain analysis, synteny and duplicated gene analysis, transcriptomic analysis, and gene co-expression analysis.

## 2. Results 

### 2.1. Identification and Characterization of NnWOX Genes

To obtain a comprehensive *WOX* gene family in lotus, three identification methods were performed (see Section 4). The orthologs of *A. thaliana WOX* genes that were downloaded from the TAIR database (https://www.arabidopsis.org/ (accessed on 1 July 2023)), genes annotated to *WOX*-related homeobox in the *Nelumbo* Genome Database (http://nelumbo.cngb.org/nelumbo/ (accessed on 1 July 2023)), and genes identified under *WOX* transcription factors in the PlantTFDB were merged into candidate *NnWOX* genes in lotus. Given the nature of conservative homeobox domain in *WOX* sequences, the candidate *NnWOX* genes were mapped to three genome databases (i.e., NCBI-CDD, SMART, and Pfam) to filter out the false-positive candidate. Finally, a total of 15 high-confidence *NnWOX* genes were identified and used for downstream analysis (Table 1). The length of these *NnWOX* proteins varied from 182 to 366 amino acids. The molecular weight gradually increased with the length of proteins. Therefore, the longest *NnWOX* protein has a maximum weight of 40,294.14 kDa and the shortest *NnWOX* protein is the lightest with a weight of 20,640.22 kDa in all *NnWOX*s. Also, the chemical and physical characteristics of *NnWOX* genes were analyzed, including theoretical PI values, instability index, aliphatic index, and grand average of hydropathicity (Table 1). Furthermore, the hydrophily and hydrophobicity of amino acids in each *NnWOX* gene sequence were analyzed (Appendix A). Our results indicate low levels of hydrophobicity and high levels of hydrophily in all *NnWOX* proteins. Except for *Nn5g30993*, 14 *NnWOX* genes have at least one N-glycosylation site (Appendix A). Interestingly, the longest *Nn1g06358* does not have the most N-glycosylation sites, but the second longest *Nn3g18264* does, which could result from the obvious differences in sequences.

### 2.2. Phylogenetic Analysis

Using the same identification pipeline of the *WOX* gene in the lotus genome, a total of nine *WOX* genes were identified in a basal dicotyledon *Amborella trichopoda*. *WOX* proteins from *A. thaliana* and *A. trichopoda* were selected to construct the phylogenetic tree (Figure 1). Combining the orthologous mapping results and the clade of the phylogenetic tree, 15 *NnWOX* and 9 *AtrWOX* genes were named according to their closest orthologs to *AtWOX*. Similar to the previous study [6], these *WOX* proteins were classified into three clades, i.e., the ancient clade (AC), intermediate clade (IC), and WUS clade (WC) (Figure 1). Of which, the WC clade has the most *NnWOX* genes and the AC clade has only two *NnWOX* genes (i.e., *NnWOX13a*, *NnWOX13b*). Additionally, three *NnWOX* genes belong to the IC clade. Notably, lotus and A. thaliana have the same number of *NnWOX* genes, but some *AtWOX* genes have no orthologs in lotus, such as *AtWOX8*, AtWOX9, and *AtWOX10*. This could be as a result of these two species undergoing different whole-genome duplication events.

### 2.3. Gene Structure Analysis of NnWOXs

The motifs, mathematical–statistical models of specific sequences, were predicted for *NnWOX* proteins. Our results showed that all *NnWOX* members have motif1 and motif2 near the carboxy terminus, which consisted of the conserved HB domain and was consistent with the specific motif in the *WOX* gene family (Figure 2). Notably, we found that *NnWOX* genes clustered in the same clade shared the same motifs. For example, *NnWOX9a*, *NnWOX9b*, and *NnWOX11* belonging to the IC clade have motif3, and motif5 was only identified in *NnWOX* genes from the AC clade (i.e., *NnWOX13a* and *NnWOX13b*) (Figure 2). A total of nine *NnWOX* genes belonging to the WC clade have the same motif4, excluding *NnWOX6a* and *NnWOX6b* that have motif6. The distinct sequence motifs in clade members were associated with their different functions. 

We analyzed the gene structures of these candidate proteins based on the locus annotation in the lotus genome for further exploration of genomic characteristics of *NnWOX* proteins. The exon number of *NnWOX* genes varied from two to four (Figure 2). *NnWOX13a* has the longest genome locus, which was significantly longer than other *NnWOX* genes (Figure 2). The untranslated regions (UTRs) were identified in *NnWOX13a* and *NnWOX4b* (Figure 2). Interestingly, we did not find similar gene structures of *NnWOX* genes belonging to the same clade, indicating large variations in lotus genome structure. Based on the well-annotated functions of *AtWOX* genes and the homologous relationships between *NnWOX* and *AtWOX* genes, we deduced that specific motifs and gene structures had a vital role in distinguishing the functions of *WOX* genes.

### 2.4. Localization and Duplicated Gene Analysis

The position of *NnWOX* genes was annotated to the pseudochromosome in the lotus genome and visualized using TBtools (Figure 3a). The 15 *NnWOX* genes were located on six lotus chromosomes, except for Chr7 and Chr8 (Figure 3a). Chr2 has the most *NnWOX* genes (six), followed by Chr1 with four *NnWOX* genes. Chr3, Chr4, and Chr5 have one *NnWOX* gene each. The physical distance between *NnWOX2* and *NnWOX3a* was the shortest compared to other *NnWOX* genes (Figure 3a). Furthermore, we found that *NnWOX* genes in the same clade of the phylogenetic tree are likely not located in the same chromosome. 

Due to lotus occurring in one ancient whole-genome duplication event [30], we investigated the duplicated type of *NnWOX* genes. According to MCScanX, a total of five whole-genome duplicated pairs of *NnWOX* genes were identified, and all these duplications belonged to the WGD/segmental duplication type. Among these five duplications, three duplicated gene pairs were in different chromosomes, including *NnWOX4a*/*NnWOX4b*, *NnWOX6a*/*NnWOX6b*, and *NnWOX13a*/*NnWOX13b* (Figure 3b). And two duplicated gene pairs were in the same chromosome, i.e., *NnWOX3a*/*NnWOX3b* and *NnWOX5a*/*NnWOX5b* (Figure 3b). Moreover, no duplicates were identified for the other five *NnWOX* genes. We speculated that the duplicates were lost due to the redundancy functions or genome variations during the evolution of lotus.

### 2.5. Gene Expression Pattern Analysis of NnWOX Genes

To gain insight into the functions of *NnWOX* genes during lotus development, we investigated their expression patterns in different tissues. The gene expression matrix of China Antique including 54 RNA-seq samples from 11 tissues was downloaded from the *Nelumbo* Genome Database [35]. The expression patterns of *NnWOX* genes are shown in Figure 4. Notably, we found that *NnWOX13a* and *NnWOX13b* have a wide tissue expression pattern, suggesting that *NnWOX* genes in the AC clade have multiple biological functions and play roles in different tissues. According to the heatmap cluster, the two *NnWOX* genes in the AC clade were clustered again, and the IC and WC members were reconstructed. This highlights the wide and vital roles of ancient *WOX* genes during lotus growth and development. Except for these two *NnWOX* genes, other *NnWOX* genes have tissue-specific bias expression patterns (Figure 4). For example, *NnWOX2*, *NnWOX3b*, *NnWOX6b*, and *NnWOX9b* have specifically high expression levels in cotyledon, and *NnWOX5b* was only expressed in root and apical meristem (Figure 4). Interestingly, few *NnWOX* genes were expressed in vegetative growth tissues, such as leaf, petiole, and rhizome. However multiple *NnWOX* genes show high expression levels in reproductive growth tissues, including stamen, carpel, and cotyledon. Therefore, *NnWOX* genes in IC and WC clades might contribute to the formation of reproductive tissues by targeting and interacting with other genes that regulate the reproductive processes.

We found distinct expression patterns between *WOX* duplicates in the IC and WC clades. *NnWOX3a* has a wider tissue expression pattern compared to *NnWOX3b*, and *NnWOX3b* has an obvious expression bias to cotyledon (Figure 4). Compared to the duplicate *NnWOX6b*, *NnWOX6a* lacks expression in the cotyledon and immature receptacles. *NnWOX9b* also has a higher expression level in seed coat and cotyledon than its duplication *NnWOX9a* (Figure 4). Due to these obviously distinct expression patterns of duplicate *WOX* genes in tissues, we speculate that different functions between duplicate *WOX* genes were formed by the neofunctionalization during the evolution of lotus.

### 2.6. Co-Expressed Gene Networks of NnWOXs

To further explore the regulatory relationships of *NnWOX* genes in lotus tissues, a weight gene co-expression network analysis (WGCNA) was conducted on the filtered gene expression matrix. Due to the WGCNA gathering genes that are highly interconnected into a module, which is significantly (*p* < 0.001) related to the traits, the genes within similar biological processes or regulatory relationships are supposed to be identified in the same module. After excluding the silenced genes among 54 tissue samples, a total of 25,432 genes were aggregated into 19 color-coded modules (Figure 5a,b). Apart from black and turquoise modules being significantly related to two tissues, most modules were significantly related to one tissue (Figure 5b). For example, MEgrey60 was significantly related to cotyledon, and MEsalmon and MEpurple were significantly related to seed coat. 

Based on the pipeline of constructing gene co-expression networks, ten *NnWOX* genes were filtered out and assigned to four modules, including MEgrey60, MEblack, MEsalmon, and MEpurple (Figure 5b,c). Among them, *NnWOX13a* and *NnWOX13b*, two AC clade members, were assigned to MEsalmon and MEpurple modules showing significant correlation with seed coat (Figure 5b,c). Four *NnWOX* genes, i.e., *NnWOX2*, *NnWOX3b*, *NnWOX9b*, and *NnWOX11*, were clustered into the MEgrey60 module and significantly associated with cotyledon (Figure 5b,c). Furthermore, *NnWOX3a*, *NnWOX4a*, *NnWOX4b*, and *NnWOX6b* were assigned to the MEblack module, which was relevant to both the cotyledon and the apical meristem (Figure 5b,c). We filtered the significantly co-expressed genes to *NnWOX* genes based on a strict threshold (see Section 4). The GO enrichment analysis of these co-expressed genes suggested biological functions enriched in reproductive and transcriptional regulatory processes, such as “floral development”, “DNA-binding transcription factor activity”, and “transcription regulator activity” (Appendix A). These co-expressed genes may be regulated by *NnWOX* and play a key role in the reproductive development of lotus.

To further explore the interaction of *NnWOX* and co-expressed genes, the gene co-expression networks for each *NnWOX* gene were graphed (Figure 5c). We found that genes in the same module have distinct co-expression patterns. For example, *NnWOX9b* and *NnWOX2* shared most co-expressed genes, but they have two co-expressed genes correlated to *NnWOX11*. Notably, although *NnWOX3b* and *NnWOX11* are clustered in the same module (i.e., MEgrey60 module), *NnWOX3b* has the only common co-expressed gene significantly correlated to *NnWOX11* (Figure 5c). However, *NnWOX* genes in the MEblack module have major common co-expressed genes, suggesting these *NnWOX* genes might regulate similar pathways in lotus (Figure 5c). The duplicated gene pairs of *NnWOX4a* and *NnWOX4b* were assigned to the same MEblack module, but different co-expressed genes were identified for these two genes, indicating their distinct co-expression network and divergent functions, respectively.

### 2.7. qRT-PCR Validation of NnWOX Genes in Co-Expression Network

According to the WGCNA networks and gene expression patterns of RNA-seq, the ten *NnWOX* genes involved in the co-expression networks exhibited a high correlation to the specific tissue. To further validate the expression level of those ten *NnWOX* genes in the tissues, we collected four tissues including cotyledon, seed coat, apical meristem, and leaf to perform qRT-PCR experiments. The leaf was considered the internal reference and the other three tissues represented the specifically expressed tissues of *NnWOX* genes. Four *NnWOX* genes were allocated to MEgrey60, which was significantly related to the cotyledon, exhibiting a higher expression level in the cotyledon compared to the other three tissues (Figure 6a,c,g,h). Our results showed a similar tendency for tissue expression patterns between qRT-PCR and RNA-seq data (Figure 6). These results indicated the high accuracy of RNA-seq and WGCNA networks and the tissue-specific expression patterns of *NnWOX* genes. 

## 3. Discussion

Transcription factors play important roles in regulating transcript initiation and gene expression during plant growth and development. As a plant-specific transcription factor, *WOX* genes play key roles in maintaining stem cell homeostasis and are involved in the growth and development of multiple tissues [40,41,42,43,44]. Considering the vital roles of *WOX* genes, a whole-genome analysis of the *WOX* family was conducted in multiple plants, including *A. thaliana* [12], wheat [44], cotton [45], melon [46], and loquat [47]. The number of *WOX* members varies in these plants. Although genome-wide identification of the *WOX* members in lotus were preliminarily studied [39], the regulatory interactions of *NnWOXs* genes need further investigation. With increasing genomics research on lotus [30,48], the *Nelumbo* Genome Database provided a great platform to investigate the whole-genome analysis of transcription factors [35]. In the present study, we also conduct genome-wide identification of the *WOX* genes in lotus according to the genomic information obtained from the published dataset. 

Although lotus has the same number of *WOX* genes compared to *A. thaliana*, some orthologs of *AtWOX* genes have not been reported in lotus, including *WOX1*, *WOX7*, *WOX8*, *WOX10*, *WOX12*, and *WOX14*. Given the basal position of lotus in the phylogenetic tree of the dicotyledon [49], the *AtWOX* genes that have no orthologs to lotus are supposed to be derived from the expansion of gene families during species evolution. Despite lotus and *A. trichopoda* occurring once during whole-genome duplication events [30,34], more WOX genes were identified in lotus than in *A. trichopoda*. This could result from the distinct amplifications and losses of WOX gene family members between these two species during plant evolution. Two *NnWOX* genes, i.e., *NnWOX13a* and *NnWOX13b*, were identified in the ancient clade; we speculated that this WGD pair remained after the ancient whole-genome duplication in lotus. The conserved homeodomain was used as a criterion to identify members of the *WOX* gene family. Additionally, we found the *NnWOX* members in each phylogenetic clade containing specific motifs and large different genomic structures, indicating the distinctive functions of different *NnWOX* genes. 

A previous study revealed that the expression patterns of genes in lotus were different in multiple tissues [32], which enabled further investigation of the tissue expression level of the *NnWOX* genes. Interestingly, we only found similar tissue expression patterns between the duplicated *NnWOX13a/b* genes in the ancient clade. Distinctive expression patterns of *NnWOX* genes within the same clade and even between duplicates were identified, suggesting different roles in regulating the development of tissues at different growth stages. The *AtrWUS* gene played an important role in the formation of stem cells and promoted vegetative-to-embryogenic transition in *A. thaliana* [50,51,52]. However, the orthologous *NnWUS* only exhibits high expression levels in the carpel before and after pollination, in the mature receptacle, but a low expression levels in the rhizome and root. This indicated that *NnWUS* likely did not regulate the stem cells but might participate in the reproductive processes in lotus. A previous study also suggested that the stable expression level of *AtWUS* in root apical meristems was the key to maintaining the cellular organization of apical meristems [53]. However, the silent expression of the *NnWUS* and high expression levels of other *NnWOX* genes in apical meristem indicated that *NnWOX* genes were the real regulators of lotus apical meristem and not *NnWUS*. We speculated that the homeostasis of stem cells regulated by *AtWUS* was a neofunctionalization in the evolution of *A. thaliana*. Notably, we found a class of *NnWOX* genes showing specific high expression levels in the cotyledon of different developmental stages, including *NnWOX2*, *NnWOX3b*, *NnWOX6b*, *NnWOX9b*, and *NnWOX11*. Due to cotyledon being the key tissue for storing and providing nutrients during seed dormancy and germination, the development of cotyledon regulated by these *NnWOX* genes was supposed to be relevant to specific phenomena that lotus seed kept active after being buried underground for hundreds of years [54]. Furthermore, previous studies revealed that *WOX* genes contributed to the formation of somatic embryogenesis in plants, such as *A. thaliana* [15], cotton [45], maize [55], and wheat [56]. Therefore, our results indicated that several *NnWOX* genes played vital roles in the development of lotus cotyledon.

Gene co-expression networks were performed to filter gene sets involved in the same biological pathways or gene pairs interacting with each other [57,58]. Due to the minimum sample size and correlation of gene modules and tissue, weight gene co-expression network analysis (WGCNA) was widely used for identifying the co-expressed gene pairs in different plant tissues or experimental treatment samples [59,60,61]. Previous transcriptomic research on lotus has constructed a gene-level network involved in only six tissues and a transcript-level co-expression network including eleven tissues using WGCNA [32,62]. In this study, we used the same pipeline of WGCNA to construct the gene co-expression networks for each *NnWOX* gene (Figure 5). Most *NnWOX* genes in the same module shared common co-expressed genes, except for *NnWOX3b* and *NnWOX11*. Two modules (MEgrey60 and MEblack) containing eight *NnWOX* genes were significantly related to the cotyledon, in line with their specific expression patterns. The GO enrichment results suggested that the significantly co-expressed genes for the ten *NnWOX* genes used in the networks were enriched in the reproductive and transcription processes. Therefore, our results indicate that *NnWOX* genes and their co-expressed genes show specific high expression levels in the cotyledon, which is likely associated with the activities taking place during the long-term storage of lotus seed.

## 4. Materials and Methods

### 4.1. Plant Materials

Wild sacred lotus was cultivated in the Xiaogan (31°33’ N, 113°25’ E), Hubei province, China. Rolled leaves, roots, seed coat, and seed cotyledon were collected from the individual plant 15 days after flowering and frozen immediately in liquid nitrogen. High-quality RNA from each sample was extracted using the RNAprep pure Plant Kit (TIANGEN) for subsequent verification experiments.

### 4.2. Identification of WOX Genes in Nelumbo nucifera

To obtain a comprehensive *WOX* gene family in lotus, we carried out three identification methods. First, a total of fifteen *WOX* genes (*AtWOX*) in the model plant *Arabidopsis* thaliana were acquired from the TAIR database (http://www.arabidopsis.org/ (accessed on 1 July 2023)), which were further used as query sequences to perform the BALSTP search with lotus protein sequences (downloaded from http://nelumbo.biocloud.net (accessed on 1 July 2023)). Only mapped results with *p*-value < 1 × 10^−5^ were retained. Second, the gene function annotation of lotus genes was also downloaded from the *Nelumbo* genome database (http://nelumbo.biocloud.net (accessed on 1 July 2023)), and the genes annotated *WUSCHEL*-related homeobox were filtered out. Third, the protein sequences of China Antique genes were mapped to the PlantTFDB database to identify different families of transcription factors. The genes identified under the *WOX* transcription factor family members were filtered. We combined the filtered genes from these three methods and defined them as candidate *NnWOX* genes. As the *WOX* gene contained a conserved HB domain, the candidate *NnWOX* genes were mapped to three public databases, including NCBI Conserved Domain Database (CDD, https://www.ncbi.nlm.nih.gov/cdd/ (accessed on 7 July 2023)), SMART database (http://smart.embl-heidelberg.de/ (accessed on 7 July 2023)), and Pfam database (https://pfam.xfam.org/ (accessed on 7 July 2023)). The candidate genes not containing the conserved HB domain region were weeded out. According to the abovementioned identification pipeline, the *WOX* genes in *Amborella trichopoda* (*AtrWOX*) were also identified, and all genomic information regarding *A. trichopoda* was obtained from the Joint Genome Institute database (https://jgi.doe.gov/ (accessed on 7 July 2023)). Based on the mapping results, the identified *NnWOX* and *AtrWOX* genes were named according to their orthology with *AtWOX* genes.

### 4.3. Chromosome Location and Structure of NnWOX Genes

The full-length protein sequence of *NnWOX*, *AtWOX*, and *AtrWOX* were aligned using ClustalW. A phylogenetic neighbor-joining tree was constructed using MEGA v7.0 software with pairwise deletion and 1000 replicates in bootstrap analysis [63]. The constructed tree was drawn and optimized using the iTOL online tools (https://itol.embl.de/ (accessed on 7 July 2023)). The physical and chemical characterizations of the *NnWOX* protein were analyzed using the Proparam tool of ExPASy (http://weB.expasy.org/protparam/ accessed on 7 July 2023)), including the number of amino acids, molecular weights, theoretical PI values, instability index, aliphatic index, and the grand average of hydropathicity. In addition, the N-glycosylation sites of *NnWOX* proteins were predicted using the NetNGlyc (https://services.healthtech.dtu.dk/services/NetNGlyc-1.0/ (accessed on 7 July 2023)), and the hydrophobicity and hydrophilicity of these protein sequences were analyzed using the ProtScale tools of ExPASy (https://web.expasy.org/protscale/ (accessed on 7 July 2023)).

### 4.4. Chromosome Location and Structure of NnWOXs

Locus information of the physical locations of *NnWOX* genes in the genomes of “China Antique” was collected from the *Nelumbo* Genome Database (http://nelumbo.cngb.org/nelumbo/ (accessed on 1 July 2023)), and the positions were drafted to chromosomes using TBtools software [64]. Because the gene annotation from the *Nelumbo* Genome Database contained multiple transcripts, the longest transcript from each gene was chosen to represent the corresponding gene. The structures of the *NnWOX* genes were displayed using TBtools software [64] to obtain the exon composition information. To identify the sequence motifs of *NnWOX* genes, the protein sequences were analyzed using MEME tools (http://meme-suite.org/meme/tools/meme (accessed on 7 July 2023)). For each sequence, the maximum of identified motifs was six.

### 4.5. Duplication Analysis of NnWOX Genes

Given that the lotus genome only occurred once during the ancient genome duplication event [30]. MSCcanX was used to further identify the duplicated type of *NnWOX* genes classifying lotus genes into five groups: singletons, WGD/segmental duplications, dispersed duplications, proximal duplications, and tandem duplications. Based on the syntenic result of duplicated genes, the duplicated *NnWOX* gene pairs were shown in a Circos plot using an RCircos package v1.2.0.

### 4.6. Gene Expression Profile and Weighted Gene Co-Expression Network Analysis

The gene expression profile of fifty-four China Antique RNA-seq among 11 tissues was downloaded from the *Nelumbo* Genome Database (http://nelumbo.biocloud.net (accessed on 1 July 2023)). Since weighted gene co-expression network analysis (WGCNA) was effective in identifying co-expressed genes and constructing gene co-expression networks, we performed WGCNA analysis based on the gene expression profile. To obtain a high-quality gene co-expression network, among fifty-four samples, genes with average FPKM (fragments per kilobase of exon model per million mapped fragments) < 0.1 were excluded first. The gene co-expression network was built using the WGCNA R-package with min-module 500. The co-expressed genes of *NnWOXs* in each module were filtered, and the top 5% of co-expressed genes were defined as significantly co-expressed genes. Subsequently, the *NnWOX* co-expression networks were constructed using Cytoscape.

### 4.7. GO Enrichment Analysis

The gene functions of lotus genes were annotated using EggNOG v5.0 with the Viridiplantae database, and the GO enrichment analysis of *NnWOX* co-expressed genes was carried out using TBtools [64]. The GO terms with *p*-value < 0.01 were defined as significantly enriched.

### 4.8. Quantitative Real-Time PCR Experiments

The specific adapters for every *NnWOX* gene were designed using Primmer5 (http://www.premierbiosoft.com/primerdesign/) (accessed on 1 February 2023) (Appendix A). To verify the expression level of *NnWOX* genes in different tissues, high-quality RNA of five lotus samples was reverse-transcribed into five cDNA samples. The qRT-PCR experiments were performed using NovoStart SYBR qPCR SuperMix plus (Novoprotein, China), and three biological replicates for each *NnWOX* gene were applied. Relative expression of *NnWOX* genes was analyzed using the 2^−∆∆Ct^ method, with the lotus *β-Actin* gene used as an internal standard gene.

## 5. Conclusions

In this study, the *WOX* gene family was identified genome-wide in lotus. Based on the phylogenetic analysis, a total of 15 identified *NnWOX* genes were divided into three groups, which were similar to the cluster described in previous studies. Gene structure analysis showed that all *NnWOX* genes contained two conserved motifs, but specific motifs were identified in different clade members, indicating the conserved sequence variations in the three groups of lotus *WOX* genes. Gene expression analysis suggested the obvious divergence of *NnWOX* genes in different lotus tissues. We constructed gene co-expressed networks for each *NnWOX* gene and found their co-expressed genes were enriched in reproductive pathways. In summary, this work provides insight into the characteristics of *NnWOX* genes, which is helpful for further research on the molecular functions in lotus.

## Figures and Tables

**Figure 1 plants-13-00720-f001:**
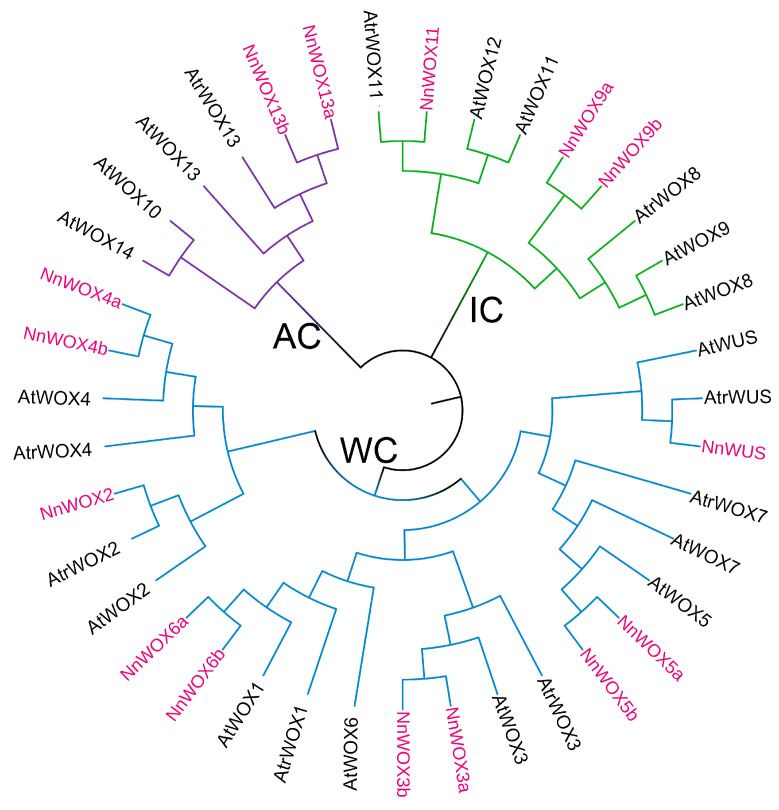
Phylogenetic tree of *WOX* proteins in *N. nucifera*, *A. thaliana*, and *A. trichopoda*. This tree could be divided into three clades, including ancient clade (AC, purple), intermediate clade (IC, green), and WUS clade (WC, blue).

**Figure 2 plants-13-00720-f002:**
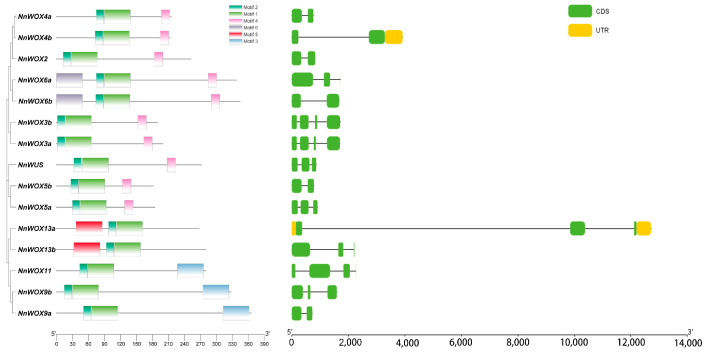
Characterizations of the identified *WOX*s in *N. nucifera*. The left panel shows the conserved motif location and the right panel shows the gene structures (black line: intron; green box: coding regions; yellow box: untranslated regions).

**Figure 3 plants-13-00720-f003:**
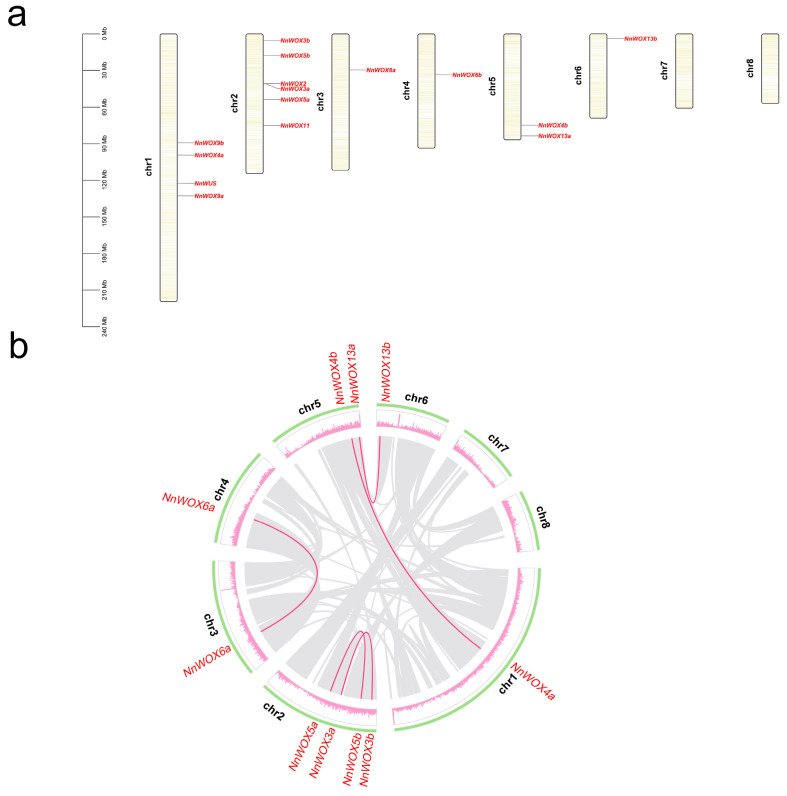
Distribution of *WOX*s in *N. nucifera* genome. (**a**) Chromosome distribution of *NnWOX*s. The scale on the left is in megabases (Mb). (**b**) Circle graph showing the location of genes in *N. nucifera* genome. The green circle represents the chromosomes, and the pink circle represents the gene density on the *N. nucifera* genome. Grey links show the collinearity regions in the lotus genome, and pink lines show the duplicated pairs of *NnWOX*.

**Figure 4 plants-13-00720-f004:**
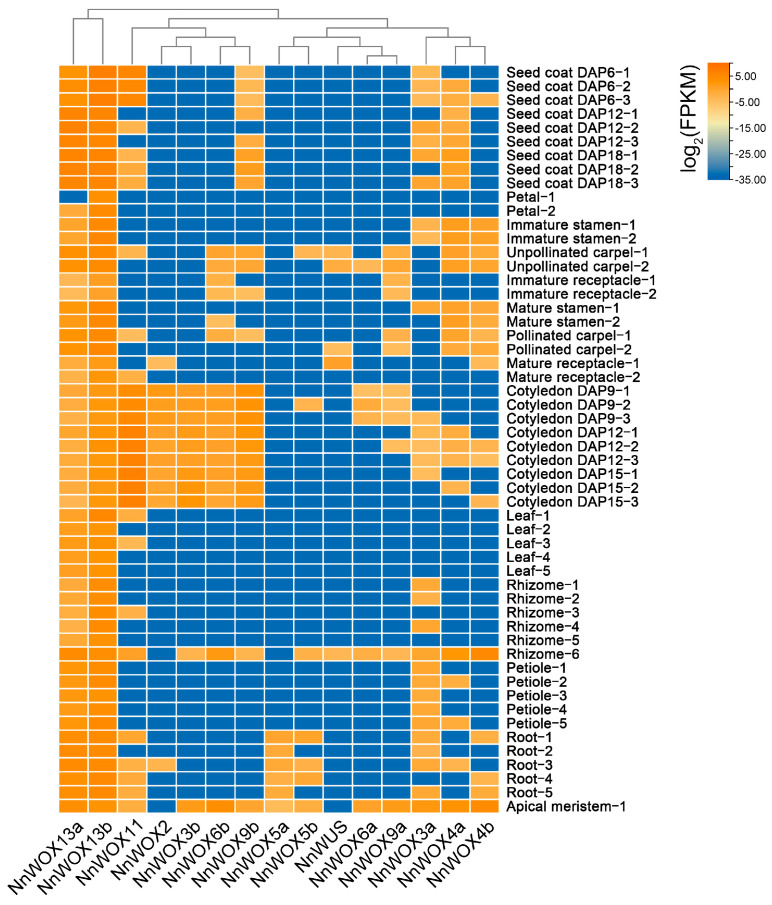
Expression patterns of *NnWOX*s in 54 tissue samples.

**Figure 5 plants-13-00720-f005:**
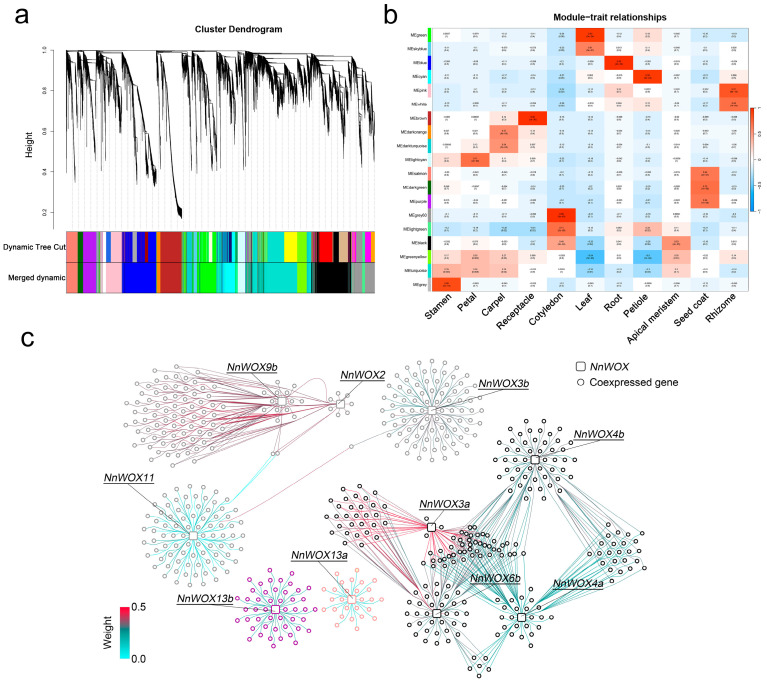
Co-expression network analysis of *NnWOX*s. (**a**) Hierarchical cluster tree and color bands indicating 19 modules. (**b**) Heatmap showing the correlation of modules and tissues. The color in each cell represents the correlation coefficient of the module in the row and tissue in the column. The value in cells shows the p-value. (**c**) Co-expression networks for each *NnWOX* gene. Squares represent the *NnWOX* genes, circles represent the co-expressed genes. The colors of genes are represented in the module, and those of the lines indicate the weight value between co-expressed genes.

**Figure 6 plants-13-00720-f006:**
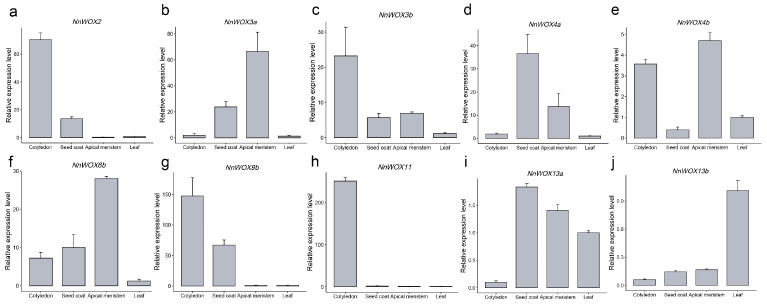
qRT-PCR verified the expression levels of *NnWOX*s in four tissues. (**a**–**j**) Ten *NnWOX* genes used in WGCNA networks that were tested. Three biological replicates were performed. The error bars show the standard error of the mean.

**Table 1 plants-13-00720-t001:** Physicochemical properties of *NnWOX* proteins in *Nelumbo nucifera*.

ID	Name	Number of Amino Acids	Molecular Weight	Theoretical PI	Instability Index	Aliphatic Index	Grand Average of Hydropathicity
Nn1g04282	NnWOX9b	328	35913.4	6.09	52.55	71.65	−0.377
Nn1g04777	NnWOX4a	216	24472.7	9.46	55.66	64.07	−0.935
Nn1g05878	NnWUS	273	30026.25	7.59	67.62	51.47	−0.794
Nn1g06358	NnWOX9a	366	40294.14	8.22	56.67	65	−0.547
Nn2g10575	NnWOX3b	190	21804.68	8.62	69.09	61.68	−0.737
Nn2g11407	NnWOX5b	182	20640.22	7.72	59.93	67.47	−0.778
Nn2g12791	NnWOX2	253	27897.2	6.76	55.71	60.91	−0.635
Nn2g12813	NnWOX3a	200	23237.49	9.07	67.16	57.15	−0.802
Nn2g13641	NnWOX5a	185	20976.68	8.7	42.02	70.05	−0.63
Nn2g14576	NnWOX11	281	30518.08	5.42	74.84	69.68	−0.289
Nn3g18264	NnWOX6a	338	38499.82	5.77	60	57.49	−0.806
Nn4g24211	NnWOX6b	345	39330.66	6.06	57.13	53.45	−0.905
Nn5g30363	NnWOX4b	214	24352.59	9.46	55.43	65.65	−0.916
Nn5g30993	NnWOX13a	268	30902.74	6.08	59.61	64.78	−0.87
Nn6g31691	NnWOX13b	280	31838.8	5.65	58.63	68.29	−0.758

## Data Availability

Data are contained within the article and Appendix A.

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
