# Peer review of "Genome-Wide Identification and Co-Expression Networks of WOX Gene Family in Nelumbo nucifera"

_plants, 2024, doi:10.3390/plants13050720_

Round 1
Reviewer 1 Report
Comments and Suggestions for Authors
Review for “Genome-wide identification and coexpression networks of WOX gene family in Nelumbo nucifera”, by Li et al.
Main points:
The paper entitled “Genome-wide identification and coexpression networks of WOX gene family in Nelumbo nucifera” by Li and collaborators reports the identification of 15 members of the Wuschel-related homebox (WOX) protein encoding genes in the sacred lotus Nelumbo nucifera. The authors have probed the N. nucifera genome using A. thaliana WOX genes family to identify the orthologous genes and further confirmed their potential identity using Pfam, SMART and NCBI-CDD databases. They analyzed the structure of the 15 NnWOX genes and their phylogeny in silico, using various bioinformatic tools. Finally they check their expression patterns in plants using RNA-seq, and produced co-expression network analysis. The work appear sound to me well executed. The paper is clearly written, except a rather confusing part (see minor point below). The work suffers, nonetheless, two major weaknesses, in my view. Firstly, except, for the RNAseq part it is based only on purely in silico evidences, and no experimental proof of the identified genes actually being coding for functional WOX genes are provided (Arabidopsis mutant complementation, for example, which is easy to do). This should have been done, I think. Secondly, a highly similar study, to say the less, have already been published in The International Journal of Molecular Sciences 2023, 24(18), 14216 (https://doi.org/10.3390/ijms241814216) by Chen and collaborators, which identified the very same genes, by a very similar approaches, but with a slightly more advanced functional characterization (GFP fusion to localize the protein in the nucleus). Moreover, besides gene expression data, Chen and collaborators also provide in silico analysis of potential targets for WOX proteins with orther FTs and regulators. This, in my view, dilutes the novelty and thus the interest of the present study for the Plant Scientist community.
Based on the above-mentioned elements I recommend the manuscript to be rejected in its current form and being resubmitted later to Plants implemented with more solid functional genomic and molecular characterization work.
Minor points:
· Introduction lines 55 to 60: you state that WOX proteins are involved in abiotic stress reposes but quote a reference [27] that deals with the role of WOx proteins in developmental process. Please clarify.
· Introduction line 66-67. I don’t understand this sentence. What do you mean by “with specific purposed of artificial domestication the cultural lotuses were divided into three groups i.e. rhizome, seed and flower lotus”. What are the “groups”? Do you mean organs? Genetic groups? Please clarify in your introduction.
· Bibliography format should be checked thoroughly throughout the reference list: issues are sometime omitted, whereas they are added in most case. Besides, article numbers are missing for MDPI journals (Plants, IJMS, …).
Comments on the Quality of English LanguageThe paper is well written and easy to read, appart from small parts of the introduction that would need rewriting (please see above, minors points) to be clarified.
Author Response
Responses:
Thank you very much for the professional opinions provided by the judges. Regarding your two concerns, my response is as follows: Firstly, the WOX gene family has a conserved domain, and the accuracy of our identification of the WOX gene family based on the well-annotated Nelumbo genome database is reliable. Secondly, although previous researchers have identified the WOX gene family in lotus, the co-expression regulatory network constructed in our article is more comprehensive. Therefore, we believe that our article also has significant reference value.work.
Minor points:
- Introduction lines 55 to 60: you state that WOX proteins are involved in abiotic stress reposes but quote a reference [27] that deals with the role of WOx proteins in developmental process. Please clarify.
Responses:
Thank you for pointing out this error. We changed the citation here.
- Introduction line 66-67. I don’t understand this sentence. What do you mean by “with specific purposed of artificial domestication the cultural lotuses were divided into three groups i.e. rhizome, seed and flower lotus”. What are the “groups”? Do you mean organs? Genetic groups? Please clarify in your introduction.
Responses:
Thank you for your question. We changed the word “groups” into “taxonomic groups” in lines 66-67.
- Bibliography format should be checked thoroughly throughout the reference list: issues are sometime omitted, whereas they are added in most case. Besides, article numbers are missing for MDPI journals (Plants, IJMS, …).
Responses:
Thank you for your advice. We double-cheaked the reference list to make it more clearly.

Reviewer 2 Report
Comments and Suggestions for Authors
In this peer-reviewed paper, genes of the WUSCHEL-related homeobox (WOX) family, which encode very important trans-factors for plant life, were identified and analysed in Nelumbo nucifera. The WOX gene family is represented by 15 members. The genome sequence was taken from the database. Numerous currently well-developed bioinformatics approaches were used to provide a comprehensive characterisation of the genes in this family. According to phylogenetic analysis, NnWOX genes were grouped into three clades. Specific motifs of NnWOX genes in different clades and divergent gene structures were found. The exon-intron structure of the genes, domain structure and various characteristics of the proteins encoded by these genes were studied. The localisation of genes on chromosomes is shown. Co-expression networks were constructed for each NnWOX gene. Transcriptome analysis of gene expression of the family was performed. The raw data for gene transcription analysis were taken from the database. Differential expression of WOX genes in different parts of the plant is shown.
Almost all the work is done by bioinformatic methods. The only experimental part is the determination of transcript levels by RT-qPCR. It does not have any biological meaning and is aimed at confirming the data of transcriptome analysis.
There are many such papers nowadays because there are many plant species and many gene families in these plants and the methodology of analysis is well developed. Prior to this work, similar analyses of WOX gene families have been performed in 5-6 plant species, which certainly made it easier for the authors to perform this study. New information has been obtained. It concerns the WOX gene family of Nelumbo nucifera and, in addition, the knowledge of the whole gene family has been enriched by obtaining new information. Thus, the obtained results somewhat expand our knowledge about the WOX gene family.
I have two minor comments on this manuscript.
1- The authors do not distinguish between organ and tissue and this is unacceptable. They refer to a tissue as a cotyledon and even a leaf. As we know, an organ is usually composed of several or many tissues. Such confusion goes in so many places in the text (I do not even highlight them), in the captions to the figures, etc. All need to be corrected.
2. I never understand when articles state the average size of proteins encoded by a gene family. It is akin to the average temperature in a hospital and devoid of any useful information.
I don't know the journal's requirement for articles that are not done experimentally, but somewhat extend existing knowledge. If the editor sees fit to accept the article, I would agree with him and request that my two comments be taken into consideration. If he does not agree to accept, given the journal's policy, that is his right. Perhaps this article would be more appropriately published in journals devoted to the characterisation of genes and genomes.
Author Response
Responses:
Thank you for your recognition of our works.
- The authors do not distinguish between organ and tissue and this is unacceptable. They refer to a tissue as a cotyledon and even a leaf. As we know, an organ is usually composed of several or many tissues. Such confusion goes in so many places in the text (I do not even highlight them), in the captions to the figures, etc. All need to be corrected.
Responses:
Thank you for your questions. Referring to the paper of Nelumbo genome dataset, here the tissue represents a tissue sample of RNA-seq. Therefore, the concept of tissue here represents sample information rather than actual plant tissue.
- I never understand when articles state the average size of proteins encoded by a gene family. It is akin to the average temperature in a hospital and devoid of any useful information.
Responses:
Your comments are accurate and professional. We deleted this descriptive sentences in line 94.
Reviewer 3 Report
Comments and Suggestions for Authors
The authors presented a comprehensive analysis of WOX genes in Nelumbo nucifera.
The article's presentation must be improved as follows:
The graphics and images have to low resolution being almost impossible to analyze: Figures 2, 3, 4, 5, 6
Line 88: the provided link it is not functional: https://nelumbo.cngb.org/nelumbo/
Line 371 MEGA software - references and software version are missing.
Line 385 - using the TBtools software - a reference or link must be provided.
Line 390 the provided link it is invalid, please change it with: https://meme-suite.org/meme/tools/meme
Lines 417: ... were designed using Primmer5 - please add a software reference or a link.
Line 422 - lotus β-Actin 422 gene - the authors should provide the primers used in reference gene amplification.
Line 432 - gene and found theicoexpressed genes - please check
Author Response
The graphics and images have to low resolution being almost impossible to analyze: Figures 2, 3, 4, 5, 6
Responses:
We submitted the high resolution figures as the individual file, please find it.
Line 88: the provided link it is not functional: https://nelumbo.cngb.org/nelumbo/
Responses:
We changed the link to “http://nelumbo.cngb.org/nelumbo/”
Line 371 MEGA software - references and software version are missing.
Responses:
We add the reference and software version to line 372.
Line 385 - using the TBtools software - a reference or link must be provided.
Responses:
We add the citation in line 386.
Line 390 the provided link it is invalid, please change it with: https://meme-suite.org/meme/tools/meme
Responses:
We change the link as you suggested.
Lines 417: ... were designed using Primmer5 - please add a software reference or a link.
Responses:
We added the link in line 419.
Line 422 - lotus β-Actin 422 gene - the authors should provide the primers used in reference gene amplification.
Responses:
We added the primer sequences in Table S1.
Line 432 - gene and found theicoexpressed genes - please check
Responses:
We revised the sentences to make it clearly in line 435.

Reviewer 4 Report
Comments and Suggestions for Authors
Dear Authors,
I have read the article „Genome-wide identification and coexpression networks of WOX gene family in Nelumbo nucifera“ where the aim focus is on in silico identification, expresssion and co-expression patterns of WOX gene family in Nelumbo nucifera.
The manuscript is very well structured. Presents relatively new information and it is with scientific relevance. The bioinformatic studies related to expression and co-expression patterns are proved by molecular methods. The abstract is informative, the introduction, results and discusion are consecutive and easy to follow.
M&M are well described, but I have some minor remarks I would like to point:
- Plant material – it will be more informative for the readers to note the age of the plants from which the samples were taken (not just young leaves, cotyledons, etc…)
- The section 4.4 is named “Plant material”. I assume this is a technical error. You should change the name of this section.
The conclusions are consistent, but it is good to mention the latin name of lotus Nelumbo nucifera to be clear.
The Figures and Tables are with good quality and present the information in an exelent way.
The references are appropriate.
I have minor remarks about the text:
- Check if the latin names are in Italic, for additional intervals, or lack of intervals. For example: line 59, line 118, line 348 - „Arabidopsis“ should be in Italic; line 72, line 353, line 384, line 386, line 401 - „Nelumbo“ in Italic;
- line 227, line 385, line 432 – interval needed;
- line 256 – „The“ with capital letter;
- line 259 – two intervals.
The quality of English language is good.
Congratulations for the very nice manuscript. I recommend minor revisions. The corrections I pointed are due to minor methodological errors and text editing.
Author Response
- Plant material – it will be more informative for the readers to note the age of the plants from which the samples were taken (not just young leaves, cotyledons, etc…)
Responses:
We added the specific development stage of plants when we collected the tissue samples.
- The section 4.4 is named “Plant material”. I assume this is a technical error. You should change the name of this section.
Responses:
We revised this error.
- Check if the latin names are in Italic, for additional intervals, or lack of intervals. For example: line 59, line 118, line 348 - „Arabidopsis“ should be in Italic; line 72, line 353, line 384, line 386, line 401 - „Nelumbo“ in Italic;
Responses:
We revised these format errors as you suggested.
- line 227, line 385, line 432 – interval needed;
Responses:
We added the interval in lines 227, 385, 432.
- line 256 – „The“ with capital letter;
Responses:
We did not find “the” that needed be capitalized at the first letter
- line 259 – two intervals.
Responses:
We revised this error in line 259.
Round 2
Reviewer 1 Report
Comments and Suggestions for Authors
I still think that the paper published by Chen and collaborator in 2023 in the International Journal of Molecular Sciences 24(18) number 14216, which largely overlaps with the findings reported here should be, at minimum cited in the manuscript, both in the introduction and in the discussion, before the paper being accepted.
Author Response
Responses:
Thank you for your professional suggestions. We have added relevant literature to the introduction and discussion sections and made modifications to the related sentences in lines 75-76 and lines 276-277. Although previous researchers have conducted similar studies, our method of constructing a co-expression network is more refined, and we have carried out RT-PCR experiments. Therefore, we maintain that our article has significant reference value.